# Dynamic Response and Molecular Chain Modifications Associated with Degradation during Mixing of Silica-Reinforced Natural Rubber Compounds

**DOI:** 10.3390/polym15010160

**Published:** 2022-12-29

**Authors:** Ammarin Kraibut, Sitisaiyidah Saiwari, Wisut Kaewsakul, Jacques W. M. Noordermeer, Kannika Sahakaro, Wilma K. Dierkes

**Affiliations:** 1Department of Rubber Technology and Polymer Science, Faculty of Science and Technology, Prince of Songkla University, Pattani 94000, Thailand; 2Sustainable Elastomer Systems, Department of Mechanics of Solids, Surfaces and Systems, Faculty of Engineering Technology, University of Twente, P.O. Box 217, 7500 AE Enschede, The Netherlands; 3Elastomer Technology and Engineering, Department of Mechanics of Solids, Surfaces and Systems, Faculty of Engineering Technology, University of Twente, P.O. Box 217, 7500 AE Enschede, The Netherlands

**Keywords:** natural rubber (NR), degradation, silica-reinforced tire compound, dynamic response, chain modification

## Abstract

Mixing silica-reinforced rubber for tire tread compounds involves high shear forces and temperatures to obtain a sufficient degree of silanization. Natural Rubber (NR) is sensitive to mastication and chemical reactions, and thus, silica–NR mixing encounters both mechanical and thermal degradation. The present work investigates the degradation phenomena during the mixing of silica-reinforced NR compounds in-depth. The Mooney stress relaxation rates, the dynamic properties with frequency sweep, a novel characterization of branch formation on NR using Δδ values acc. Booij and van Gurp-Palmen plots, together, indicate two major competitive reactions taking place: chain scission or degradation and preliminary cross-linking or branch formation. For masticated pure NR and gum compounds, the viscous responses increase, and the changes in all parameters indicate the dominance of chain scission with increasing dump temperature. It causes molecular weight decrease, broader molecular weight distribution, and branched structures. Different behavior is observed for silica-filled NR compounds in which both physical and chemical cross-links are promoted by silanization and coupling reactions. At high dump temperatures above 150 °C, the results indicate a significant increase in branching due to preliminary cross-linking. These molecular chain modifications that cause network heterogeneity deteriorate the mechanical properties of resulting vulcanizates.

## 1. Introduction

Three decades have passed since the first patent filed by Michelin [1] in the 1990s for silica-reinforced synthetic Solution Styrene Butadiene Rubber/Butadiene Rubber (SSBR/BR) blend compounds for low rolling resistance tires. Silica-reinforced tire tread compounds have been increasingly used, driven by the need for fuel-saving or more energy efficiency. A major issue in utilizing silica in tire rubbers arises from the polar nature of silica itself which negatively affects compatibility with nonpolar rubbers, i.e., SBR, BR, and NR. Therefore, a vast number of studies on silica-reinforced rubber compounds have been focused on the enhancement of compatibility and good dispersion of silica in the rubber matrix. To achieve optimum reinforcement, bifunctional organosilanes such as bis-(3-TriEthoxySilylPropyl) Tetrasulfide (TESPT) and bis-(3-TriEthoxySilylPropyl)-Disulfide (TESPD) are commonly used as coupling agents that can functionalize the silica surface via a condensation reaction with the silanol groups, so-called “silanization”. To ensure the occurrence of the silanization reaction, it needs a high temperature during mixing, approximately in the range of 135 °C–155 °C [2,3,4,5,6,7]. On the other hand, for NR, during these high temperatures of silanization, thermo-oxidative degradation can be simultaneously induced due to the heat and oxygen in the atmosphere [8,9].

As early as 1938, it was reported that thermo-oxidative degradation deteriorated the mechanical properties of rubber vulcanizates [10]. A study of thermo-oxidative degradation of unfilled and carbon black-filled NR reported an increase in oxidation rate with increasing temperature, which resulted in a decline of tensile strength and elongation at break [11]. Thermo-oxidative degradation of NR was reported when mixing was performed at high temperatures, i.e., 200 °C, and it was found to increase with higher mixing temperatures [12]. Narathichat et al. [13] observed degradation of NR that was compounded in an internal mixer at high temperatures of 160 °C and 180 °C for a prolonged mixing time of 10 minutes. Higher mixing temperatures led to a decreasing trend of tensile strength in Thermoplastic Polyurethane (TPU)/NR blends [14]. For silica–rubber mixing, the effects of mixing temperature on the silanization reaction using 3-OctanoylThio-1-PropylTriEthoxySilane (OTPTES) and TESPT as silane coupling agents were reported. High mixing temperatures over 150 °C promoted the silane–silica coupling efficiency but deteriorated the mechanical properties of NR, such as modulus and tensile strength [6,8,15,16]. A similar trend was found in silica-filled NR using a secondary filler [17] and when using Hydroxy-Terminated NR (HTNR) as silica interfacial modifier [18], where the tensile strength, moduli at 100% and 300% elongation decreased with high dump temperature, especially over 160 °C. However, there was neither clear evidence nor an in-depth investigation to elaborate on such NR degradation during silica/rubber mixing. Therefore, for a better understanding of the rubber degradation phenomenon during silica–rubber mixing, an in-depth study must be carried out. By understanding the phenomenon and its causes, it is expected that better control of mixing can be planned to minimize the degradation and enhance the final properties, especially in the case of NR, which is prone to degradation by various factors.

In the present work, a silica-reinforced NR-based tire tread compound is investigated with a focus on its thermo-oxidative degradation caused by mixing. Due to various components being mixed into NR and the silica–silane–rubber mixing conditions needing high temperature as well as extensive shearing, competitive reactions between the silanization and degradation coincidently occur. This may lead to higher, lower, or even balanced compound properties and complicates the judgment of NR degradation. Therefore, the effect of mixing conditions and some key ingredients of silica-reinforced NR-based compounds on rubber degradation needs an in-depth investigation. The present work reports the influence of mixing conditions, in particular dump temperature, and mimics prolonged mixing on the degradation behavior of both unfilled and silica-filled NR compounds. For reference, masticated pure NR without any additive subjected to similar temperature and shearing residence time is also investigated. Degradation of the NR is characterized by various techniques to monitor structural changes, viscoelastic responses, and changes in molecular weight, with special emphasis on long-chain branch formation using the Δδ (frequency dependence of the phase angle δ) according to Booij [19] and van Gurp-Palmen [20]. In addition, the compound and vulcanizates properties are evaluated with a focus on tire tread-related performance.

## 2. Experimental

### 2.1. Materials 

Standard Malaysia Natural Rubber 10 (SMR10), the NR grade most commonly applied in tire treads, was provided by WEBER & SCHAER GmbH & Co. KG, Hamburg, Germany. Silica ULTRASIL 7005 with CTAB (Cetyl-Trimethyl-Ammonium Bromide) and BET (Brunauer-Emmet-Teller) specific surface areas of 171 and 190 m^2^/g, respectively, and silane coupling agent bis-(3-triethoxysilylpropyl)-disulfide (TESPD) were obtained from Evonik, Germany. The other ingredients are Treated Distillate Aromatic Extract oil (TDAE oil) (Vivatec 500) (Hansen & Rosenthal, Germany), DiPhenyl Guanidine (DPG), N-Cyclohexyl-2-Benzothiazyl Sulfenamide (CBS), 2,2,4-TriMethyl-1,2-dihydroQuinoline (TMQ), N-(1,3-dimethylbutyl)-N′-Phenyl-p-PhenyleneDiamine (6PPD) (all from Lanxess Rhein Chemie Gmbh, Cologne, Germany), and Zinc Oxide (ZnO), Stearic Acid, and Sulfur (all from Merck KGaA, Darmstadt, Germany).

### 2.2. Preparation of Masticated Pure NR and Rubber Compounds 

Three different sets of rubber compounds, including masticated pure NR, a gum/un-reinforced/unfilled compound, and a silica-filled NR mix, were prepared using the formulations shown in Table 1. The filled NR formulation was based on truck tire tread compounds according to the previous work of Kaewsakul et al., 2012; ref. [6]. The gum NR compound was derived from there, and the masticated pure NR was meant as a reference for raw rubber.

The mastication of NR and mixing of all rubber compounds were performed using an internal mixer with a mixing chamber of 50 cm^3^ (Brabender GmbH & Co. KG, Duisburg, Germany) and a fill factor of 0.7. 

#### 2.2.1. Masticated Pure NR

Masticated NR was prepared by using only the first step of the mixing procedures as given in Table 2, omitting the addition of any other ingredients. Due to the lower shear forces generated inside the mixer chamber in the absence of filler, less heat is generated in the rubber. To mimic final temperatures similar to the values obtained for the silica-filled rubber compound, pure NR was masticated under different mixer temperature settings (i.e., 65 °C, 90 °C, 110 °C, and 145 °C) compared to those applied to the gum and filled NR compounds.

#### 2.2.2. Gum Compounds

Unfilled NR compounds were prepared in two steps following the mixing procedures applied for silica-filled NR, as shown in Table 2. Similar to the case of masticated pure NR in Section 2.2.1, in the absence of solid filler, the dump temperature of the unfilled compound after the first mixing step is expected to be lower than that of silica-filled NR. Therefore, the initial mixer temperature setting (i.e., 70 °C, 95 °C, 115 °C, and 140 °C) was adjusted to reach dump temperatures close to what was found for the silica-filled NR compound.

#### 2.2.3. Silica-Filled NR Compounds

The silica-filled NR compounds were mixed using varied initial mixer temperature settings (i.e., 40 °C, 65 °C, 90 °C, and 120 °C), and the rotor speed was adjusted between 50 and 80 rpm to achieve various dump temperatures in a range of 100–170 °C. The mixing procedures are described in Table 2. In the first step, NR was initially masticated for 1 min, then half of the silica and all amounts of silane were added. The second half of silica, together with process oil, stearic acid, TMQ, and 6PPD, was added after 2:30 min. The mixing continued to the total mixing time of 6:45 min. The dump temperatures of the compounds were then measured by means of a hand-held thermocouple sensor stuck into the compounds. The second mixing step was to prepare a productive compound by adding ZnO, DPG, CBS, and sulfur after kneading the masterbatch for 1 min at a rotor speed of 30 rpm, fill factor of 0.7, and an initial mixer temperature setting of 70 °C. The mixing was continued for another 4 mins, then discharged and finally sheeted out on a two-roll mill. 

### 2.3. Testing of Unvulcanized Compounds

#### 2.3.1. Mooney Viscosity and Mooney Stress Relaxation 

Mooney viscosity and Mooney stress relaxation were tested after 24 h of resting using a Mooney viscometer (MV200VS, Alpha Technologies) according to ASTM D1646. ML 1+4 (100 °C) is reported, and Mooney stress relaxation was tested for 60 seconds after the rotor stopped. The stress relaxation rate (a) of rubber is described by using a power-law model of material response as shown in Equation (1):M = k(t)^a^
(1)
where: M = Mooney units (torque);

t = relaxation time (s);k = a constant equal to the torque in Mooney units 1 s after the disk is stopped;a = exponent that determines the rate of stress relaxation.

Such power law model equation can be expressed in a log-log expression as given in Equation (2):log M = a (log t) + log k(2)

By plotting log M against log t, the slope a can be determined defined as the rate of stress relaxation. This stress relaxation behavior of rubber compounds has contributions from both elastic and viscous responses. A delayed rate of relaxation indicates a higher elastic component in the overall response, whereas a swift rate of relaxation is due to a higher viscous component. The rate of stress relaxation has been found to correlate with rubber structural characteristics such as molecular weight distribution and gel content [22,23].

#### 2.3.2. Dynamic Properties as Functions of Frequency to Characterize Long-Chain Branching

Dynamic properties of all uncured compounds were characterized using a Rubber Process Analyzer (RPA) from TA Instruments. The oscillating frequency was set in the range of 0.1 to 20 Hz for a constant strain of 5% and 100 °C. At this strain magnitude, linear viscoelasticity of the material was observed. The storage (G′) modulus, loss (G″) shear modulus, and loss tangent, tan δ = G″/G′, were analyzed as a function of frequency. The change of phase angle δ with frequency according to Booij [19] and against G* according to van Gurp-Palmen [20] was employed to characterize long-chain branch formation, a new approach for rubbers and NR in particular.

#### 2.3.3. Cure Properties 

Cure characteristics of the compounds were recorded using a Rubber Process Analyzer (RPA) from TA Instruments at 150 °C for 30 mins according to ASTM D5289.

### 2.4. Testing of Vulcanized Rubbers

The compounds were vulcanized to their optimum cure times (tc90) using a Wickert Laboratory Press (WLP1600, Wickert Maschinenbau GmbH, Landau, Germany) at 150 °C and 100 bars into 2 mm thick sheets. Type 2 dumbbell test specimens were die-cut from the press-cured sheets, and tensile tests were carried out with a Zwick tensile tester (model Z1.0/TH1S) at a cross-head speed of 500 mm/min according to ASTM D412. The moduli at 100% and 300% strain, tensile strength, and elongation at break were recorded.

## 3. Results and Discussion

The final mix temperature reached during mixing, as represented by the dump temperature, is of paramount importance for silica-filled rubber while using a silane coupling agent. This is in order for the silanization reaction to proceed fully, whereby the polar silica surface is chemically shielded by the polar side of the coupling agent. The other side allows for participation in the later vulcanization so as to achieve chemical coupling of the silica to rubber chains. For optimal silanization, temperatures around 140 °C are commonly required. A higher dump temperature does not lead to much further improvement. However, this high dump temperature of 140 °C poses a major challenge to the mixer operator because prior data already showed that this is on the verge of NR breakdown [6], which is the essence of the present study.

### 3.1. Mooney Viscosity and Mooney Stress Relaxation

Mooney viscosities and the stress relaxation rates of pure NR and non-productive gum and filled compounds subjected to different dump temperatures are shown in Figure 1 and Figure 2. For the masticated pure NR and gum compounds, a gradual decrease in Mooney viscosity and rising rate of stress relaxation are observed with increasing dump temperature. A lower Mooney viscosity may be associated with the shortening of the rubber’s molecular chains induced by the shearing action in the mixer. Meanwhile, as the Mooney stress relaxation behavior is a combination of both elastic and viscous responses, a higher relaxation rate indicates an increase in the viscous/elastic ratio. In polymer rheology, stress relaxation is determined by a combination of both elastic and viscous responses, where a higher relaxation rate indicates an increase in viscous/elastic ratio: more liquid-like behavior than solid-like. An increase in this viscous/elastic ratio and, therefore, more liquid-like behavior is another sign of molecular mass decrease, i.e., a rising degree of degradation due to chain scission. Therefore, the results indicate more chain scission and thermal oxidative degradation of the unstabilized pure NR molecules at higher mixing temperatures. This degradation of NR during mixing at high temperatures is well-known and has been demonstrated previously, such as in the works of Narathichat et al. [13], Kaewsakul et al. [6], Sarkawi et al. [16], Sae-oui et al. [24], and Sattayanurak et al. [17]. 

Comparing the masticated/unstabilized pure NR and gum compounds, the changes in Mooney viscosity and the stress relaxation rate with dump temperatures show the same trend. The Mooney viscosity of pure NR is higher than that of the gum in the low dump temperature range, i.e., 120–150 °C, but this difference is reversed when the dump temperature exceeds 150 °C. At lower temperatures, the presence of stearic acid and antioxidants, i.e., TMQ and 6PPD, facilitates chain movement and thus lowers the viscosity. At very high temperatures, such as 160 °C, when degradation becomes dominant, the presence of antioxidants in the gum compound helps to suppress the degradation to some extent and thus prevents a drastic change, as is the case for pure NR. The present results confirm that the mixing temperature of NR should not exceed 150 °C, in agreement with the range of silica–NR dump temperatures suggested by Kaewsakul et al. [6].

For silica-filled compounds, the highest Mooney viscosities are observed due to the presence of solid fillers and the creation of filler–rubber interactions via the silanization and coupling reactions of the silane used. The Mooney stress relaxation rate is slowest, indicating the greatest extent of elastic response compared to the gum compound and masticated pure NR for the same reason of the filler–rubber interactions. Both results confirm an optimum dump temperature of around 140 °C, as previously reported by Kaewsakul et al. [6].

With increasing dump temperature, different trends are observed for the unfilled gum and pure NR compounds compared to the filled one. The Mooney viscosity first slightly rises and then strongly decreases when the dump temperature surpasses 140–150 °C. The slight increase is the result of preliminary branch formation, but later on, overtaken by molecular breakdown leading to a strong decrease in Mooney viscosity. These phenomena are accompanied by a steady decrease in the Mooney stress relaxation rate. This means a steady increase in elastic response over viscous related to the branch formation mentioned before.

The chain scission of NR during mixing can be attributed to both high-temperature mastication by the shearing action and oxidative degradation. These are shown in Figure 3. The molecular chain scission results in radicals that can readily combine with oxygen or with other polymer radicals [25,26,27], as shown in Figure 3a. In the presence of oxygen and accelerated by high temperature, an intermediate cyclic structure can be formed in NR, in which the oxygen–oxygen bonds are broken, leading to chain scission and shorter polymer chains with carbonyl end-groups [28,29], as shown in Figure 3b. The shorter chains with lower molecular weight increase the mobility, and so decrease the Mooney viscosity. At the same time, the elastic response is increased, on the other hand, as reflected by the decreased Mooney stress relaxation rate due to primarily branch formation.

### 3.2. Dynamic Properties as a Function of Frequency

Figure 4 shows the G′ and G″ as a function of frequency of uncured-masticated pure NR, gum, and filled NR prepared at four different dump temperatures. The silica-filled NR compounds clearly exhibit different trends of the moduli with frequency and also with the dump temperatures compared to those of the masticated pure NR and gum samples. For the same type of rubber or compound subjected to different mixing conditions, the changes in moduli may be related to molecular chain modifications developed in the material, as highlighted before.

Generally, the evolution of the moduli again reflects the competition between chain scission and chain recombination/branch formation [30], respectively, and also some cross-linking due to the coupling agent reacting for the silica-filled compounds. Chain scission increases molecular mobility and thus enhances viscous behavior over elastic. On the other side, cross-linking and/or interactions enhance the elastic response. For the masticated pure NR and uncured gum compounds, the changes in moduli with increasing dump temperature are, therefore, an indication of primarily chain scission/degradation. The slight increase in moduli going from approximately 124 °C to 140 °C dump temperatures may be related to the onset of branch formation. The lower moduli for the gum compound relative to the pure NR are again the result of the presence of stearic acid and antioxidants acting slightly as plasticizers. The lesser degree of decrease in the moduli with increasing dump temperature for the gum compound relative to pure NR is obviously also related to the effect of antioxidants: Figure 4b vs. Figure 4a. This all corresponds with the findings of the Mooney viscosity and Mooney stress relaxation: Figure 2.

A new way of looking at these data is taking the phase angle delta (arctan (G″/G′)) in the representation of Booij [19] as function of frequency or as function of the absolute modulus acc. to van Gurp-Palmen or vGP [20]. As seen in Figure 5 and Figure 6, the dynamic responses at varying frequencies of the filled NR compounds are totally different from those of the reference unfilled ones. 

Booij’s concept was originally based on the principle that at low frequency, e.g., 0.01 Hz or 0.06 rad/s, the phase angle is highly sensitive to branching, tending to 90° for prevailing viscous behavior at no long chain branching, and to low value for prevailing elastic behavior. At high frequency, e.g., 16 Hz or 100 rad/s, the phase angle is primarily determined by short-range segmental motions in the polymers, irrespective of whether they are branched or not. The difference is given as:Δδ = δ(0.01) − δ(16) (3)

The lower the Δδ, the more the branching. The Δδ values listed in Figure 5a,b indeed signify a small tendency towards branching with increasing dump temperature for these still unfilled compounds. 

According to the vGP approach, a refinement of Booij’s concept, the shift of a whole δ-G* curve towards smaller phase angle values implies a higher polydispersity and higher long chain branching in polymers. One may employ the area under the curve, called the van Gurp Area, as a parameter to correlate with these two molecular characteristics of polymers [31]. In this context, the shift of the vGP curves is taken into consideration. Figure 6a,b illustrate that pure NR and gum compound mixed at the highest dump temperature, i.e., ca. 165 °C, exhibit the largest shift of the vGP curves towards lower phase angle values, particularly at low complex modulus or low-frequency regions. This indicates a more pronounced development of chain branching in these two samples. Overall, shifting tendencies of the curves towards lower phase angle values are visible with increasing dump temperature. This result is in good agreement with Δδ values derived from Booij’s approach. The findings from these two techniques confirm the increase in branch formation in pure NR and unfilled compounds with increasing mixing temperature. It is worth noting that both techniques give an indication of the extent of chain branching irrespective of the different molecular weights of a polymer, which is consistent with previous results reported by Hatzikiriakos [31]. Interestingly, when scrutinizing the Mooney viscosities of pure NR and gum compounds (Figure 2), a sign of branch formation can be detected by the increased viscosities of the ones mixed from ca. 120 to 140 °C, but this effect is then overshadowed by a substantial reduction in molecular weight with further increased dump temperature due to chain scission. Nevertheless, Mooney stress relaxation seems to be a good tool to characterize this branching degree since the results are in line with Booij’s and vGP’s correlations. Therefore, the levels of branch formation in this investigated series can systematically be verified by these techniques.

The shapes of the Booij and vGP curves of uncured silica-filled NR compounds look completely different, Figure 5c and Figure 6c, when compared with pure NR and unfilled ones, Figure 5a,b and Figure 6a,b. This is because the filled compounds contain silica and a silane coupling agent, which preliminarily create lightly molecular cross-links forming the beginning of a three-dimensional network inside the compounds through silanization and coupling reactions during mixing. These key reactions strongly depend on mixing temperature [6,16,17]. As can be seen from the results, increasing dump temperature significantly shifts the Booij and vGP curves towards lower phase angle values, reflecting the greater degree of branch formation, in this respect, a ‘silica–silane–rubber three-dimensional network’. The influence of silica/silane clearly overrules the effects of chain scission and branch formation in pure NR and its gums, as also observed in the Mooney results (Figure 2).

### 3.3. Cure characteristics

The cure characteristics of gum and silica-filled NR compounds as measured by RPA at 150 °C are shown in Figure 7. As to the gum compounds, the dump temperatures of 151 °C and 165 °C result in similar cure properties, which only show a slight reduction in minimum cure torques compared to the ones mixed with lower dump temperatures. The minimum torques usually relate to the Mooney viscosities, as shown in Figure 1, and may be attributed to the presence of shorter chains caused by the degradation. In the case of the silica-filled NR compounds, the minimum torques of all mixes with dump temperatures of 123 °C to 165 °C are at a similar level. The maximum cure torque is clearly lower for the dump temperature of 165 °C. In this silica–silane-filled NR system, the use of a dump temperature in the range of 135–150 °C has been reported to provide optimum properties [6] due to an optimal silanization reaction. The use of a dump temperature exceeding 160 °C provides a drastic increase in rubber degradation, which causes chain scission, leading to reduced molecular weight and, therefore, lowered maximum cure torque. In addition, reversion is observed, another sign of molecular weight reduction. The compound mixed at 123 °C tends towards higher curing speed, attributed to a higher amount of reactive sites for the sulfur vulcanization reaction compared to the compounds mixed at higher temperatures. This is due to the fact that the silanization and coupling reactions at this low dump temperature were still in a very preliminary state [2,3,6,16].

### 3.4. Tensile Properties

Figure 8 shows the 300% modulus, tensile modulus or tensile strength, and elongation at break of gum and filled NR vulcanizates prepared at different dump temperatures. For the gum vulcanizates, a marginal change of tensile modulus at 300% is observed, but tensile strength and elongation at break show an optimum at 140–150 °C and then decrease [6]. The shorter chains due to chain scission and the chain modifications caused by polymer breakdown generate radicals that form a new network as well as branched structures, leading to a drastic deterioration in the ultimate strength properties because of network imperfections.

For the silica-filled NR vulcanizates, with increasing dump temperature, a slight increase in the 300% modulus is observed. Tensile strength shows a similar optimum at 140–150 °C, Figure 8, but thereafter reduces sharply. The elongation at break is more or less constant up to 150 °C and from there on decreases as well. The increase in the dump temperature enhances the silanization and, thus, filler–rubber–filler interactions. Again, as reported previously by Kaewsakul et al. [6], the optimum silanization in silica–silane-filled NR was in the range of 135–150 °C, and with dump temperature above 150 °C, degradation of the NR molecules became dominant. The present work confirms the severe drop in the ultimate strength and elongation at break when the dump temperature of silica-filled NR is too high, 165 °C in this case. At excessive mixing temperatures, degradation by chain scission gives, in addition to shorter chains, also the formation of branched structures, which partially compensate for each other. However, overall this leads to an improper network structure and, therefore, inferior properties. 

The approach of future research is to investigate the effect of processing and aging stabilizers to prevent the degradation of natural rubber, which is more sensitive towards molecular breakdown and rearrangement than synthetic alternatives.

## 4. Conclusions

The present study demonstrates the intricate balance needed during the mixing of silica/NR compounds with coupling agents in order (1) to achieve appropriate chemical bonding between the two and (2) to prevent excessive polymer degradation. The results demonstrate that a dump temperature of ±140 °C is required for sufficient silanization. However, at that temperature, degradation starts to play a significant role. For pure NR and gum compounds, the changes in properties are mainly the result of chain scission or degradation and a compensating amount of long-chain branch formation. The analysis of the long-chain branching with the Δδ gives supporting evidence for the generation of branched structures as a result of increasing dump temperature. A different behavior is observed for the silica-filled NR compounds, whose properties show an overruling effect of the rubber–silica interactions/cross-linking on the elastic response. The changes in properties in relation to the dump temperatures demonstrate the existence of competitive reactions between the degradation of the pure rubber and cross-linking due to rubber–silica interaction. The latter involves both physical and chemical cross-links promoted by silanization and coupling reactions. The chain modifications during mixing by either chain scission, chain recombination, and long-chain branch formation clearly influence the mechanical properties of vulcanizates. The overall results mark the intricate balance needed to be attained by the mixer operator between proper silanization with as little as possible degradation of NR in order to achieve optimal properties.

## Figures and Tables

**Figure 1 polymers-15-00160-f001:**
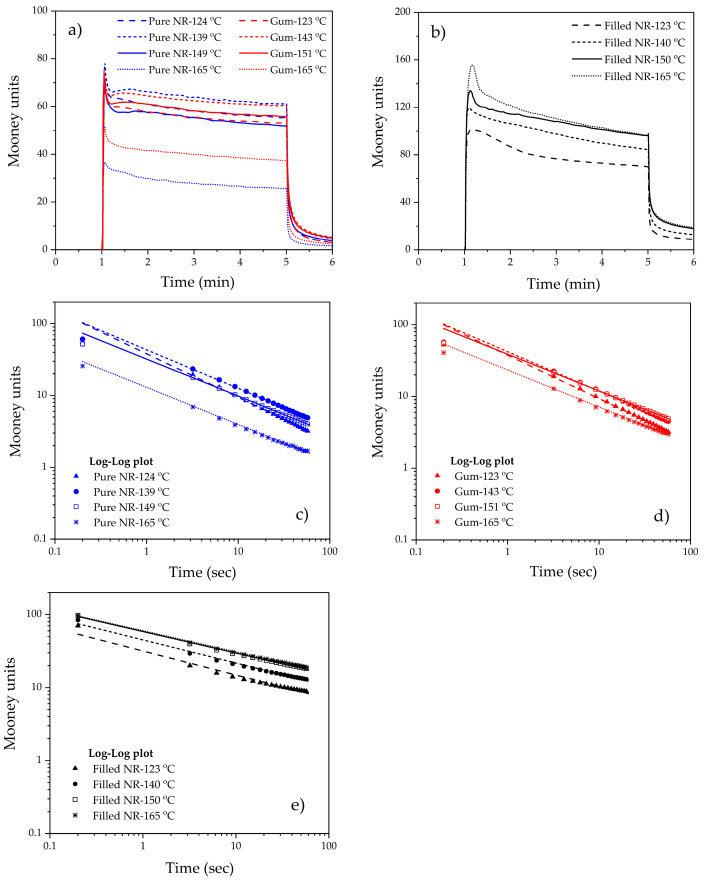
Mooney curves (**a**,**b**) and double logarithmic plots of Mooney stress relaxation (**c**–**e**) of masticated pure NR, unfilled, and filled NR compounds subjected to various dump temperatures.

**Figure 2 polymers-15-00160-f002:**
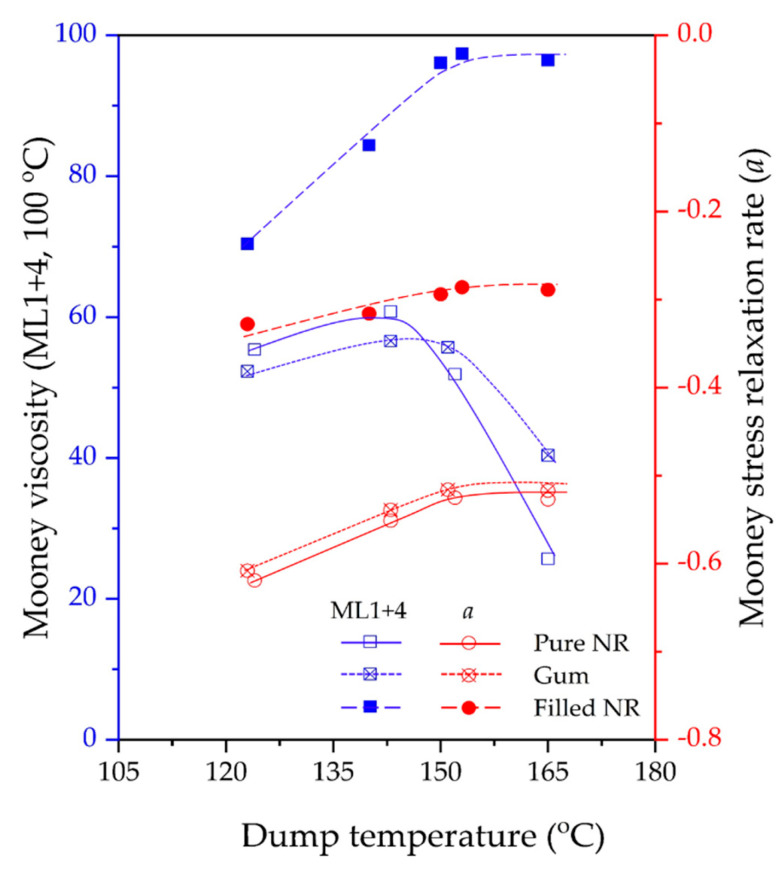
Mooney stress relaxation rate (*a*) and Mooney viscosity, ML1+4 (100 °C) of the masticated pure NR, gum/unfilled, and filled NR compounds subjected to various dump temperatures. Note the negative representation of the Mooney stress relaxation rate (*a*).

**Figure 3 polymers-15-00160-f003:**
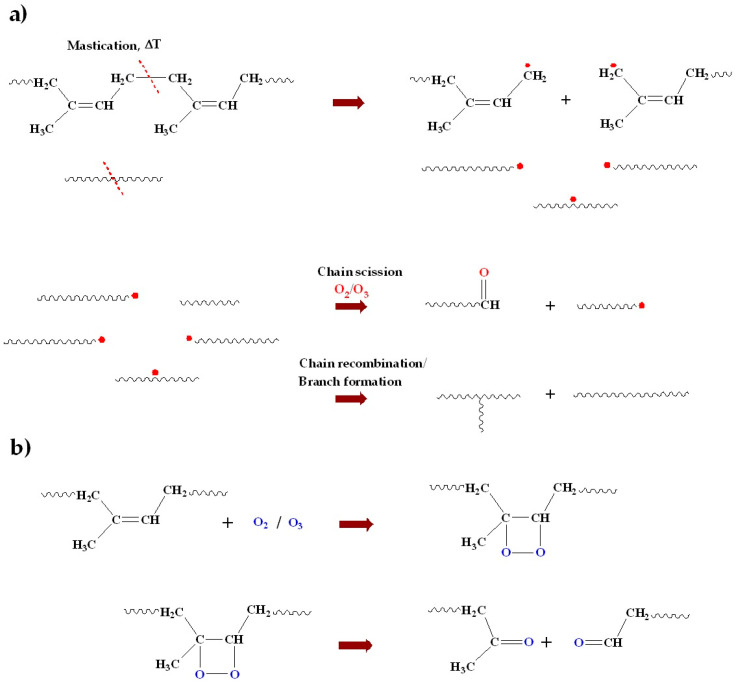
Mechanism of mechanical or thermal degradation (**a**); and oxidative degradation of NR (**b**).

**Figure 4 polymers-15-00160-f004:**
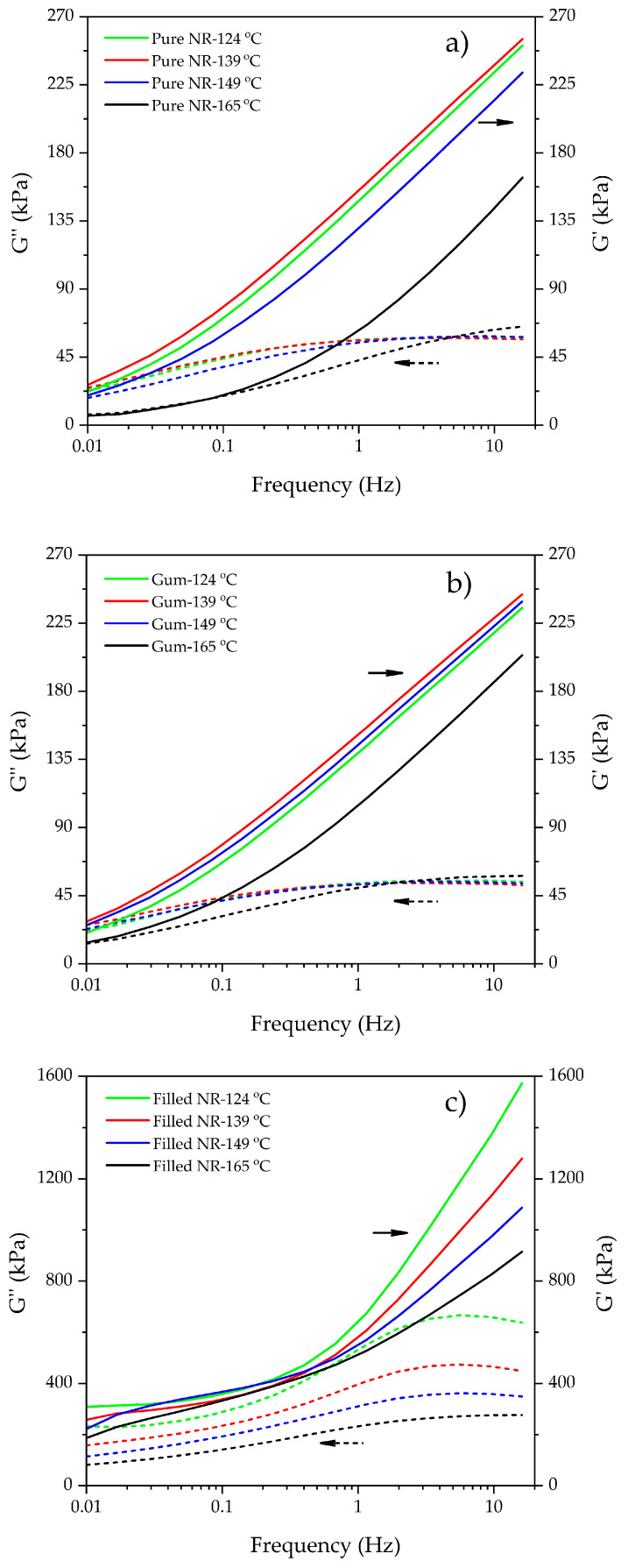
Storage and loss moduli as function of frequency for the masticated pure NR (**a**), gum (**b**), and filled NR (**c**) subjected to different dump temperatures.

**Figure 5 polymers-15-00160-f005:**
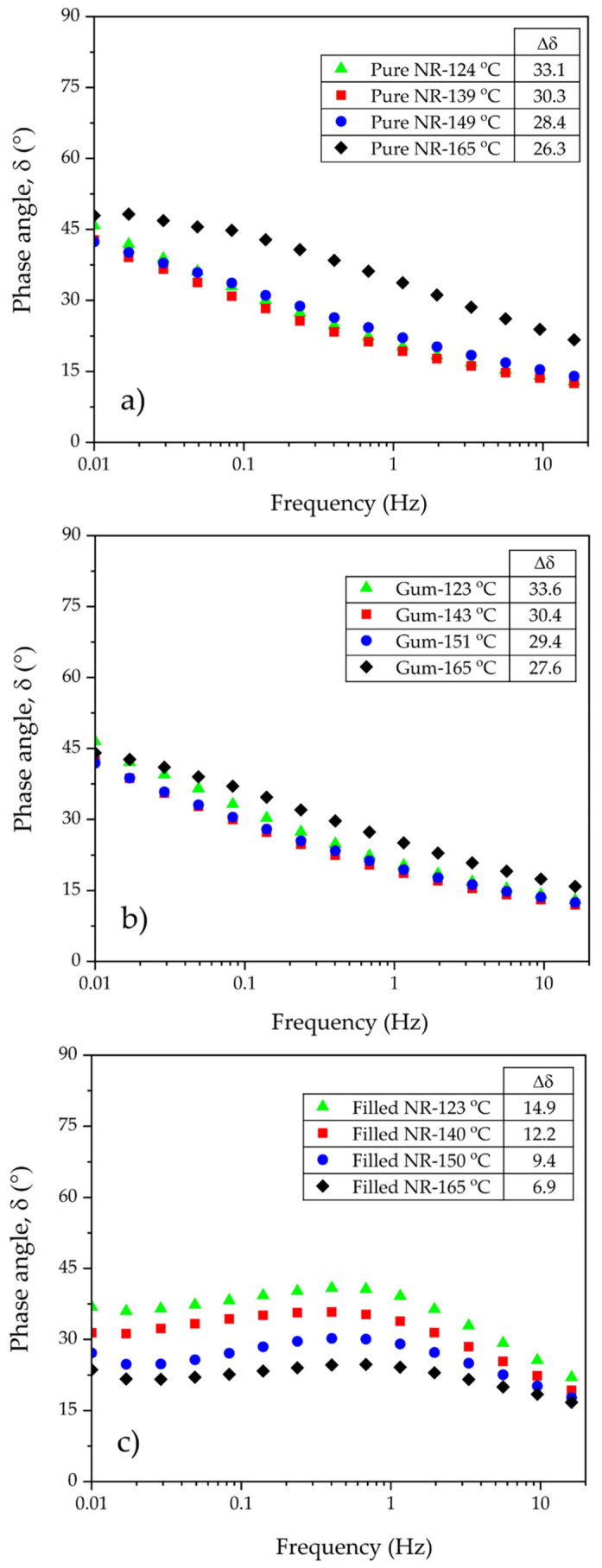
Delta or viscoelastic phase angle as a function of frequency for the pure (**a**), gum (**b**), and filled NR (**c**) compounds subjected to different dump temperatures.

**Figure 6 polymers-15-00160-f006:**
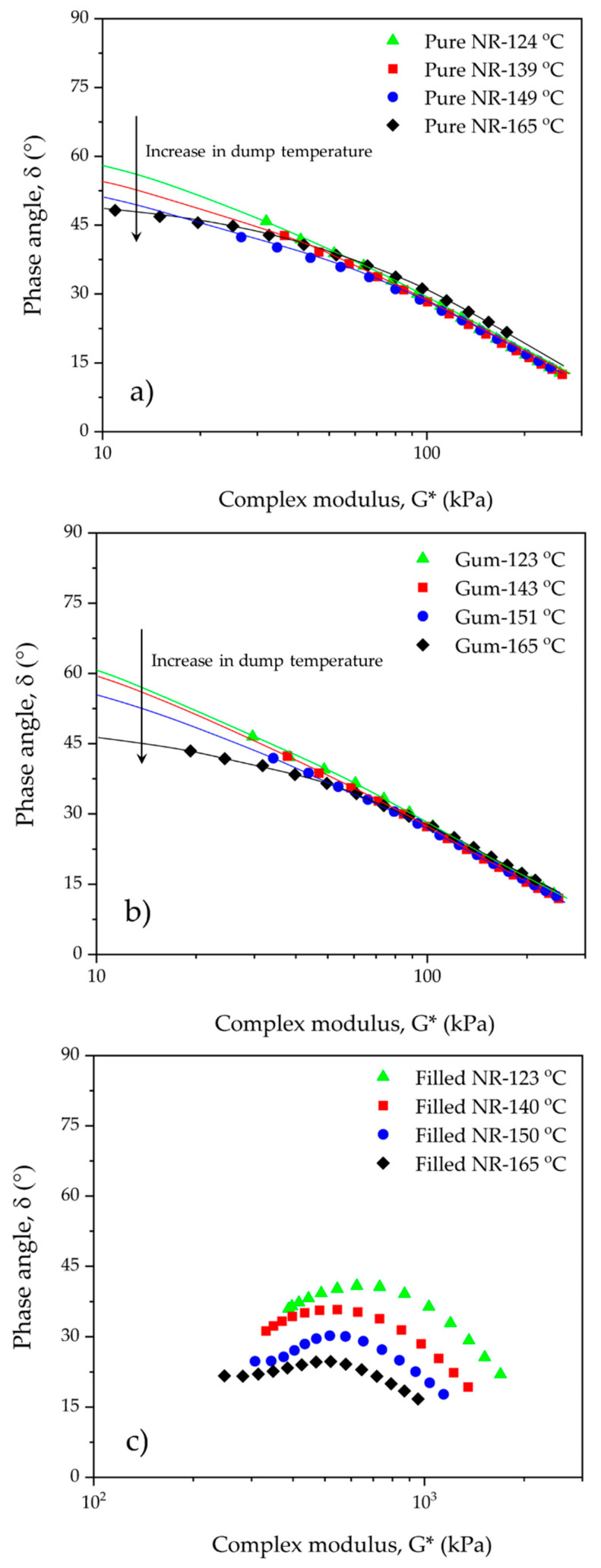
Delta or phase angle as a function of complex modulus (G*) for the pure (**a**), gum (**b**), and filled NR (**c**) compounds subjected to different dump temperatures.

**Figure 7 polymers-15-00160-f007:**
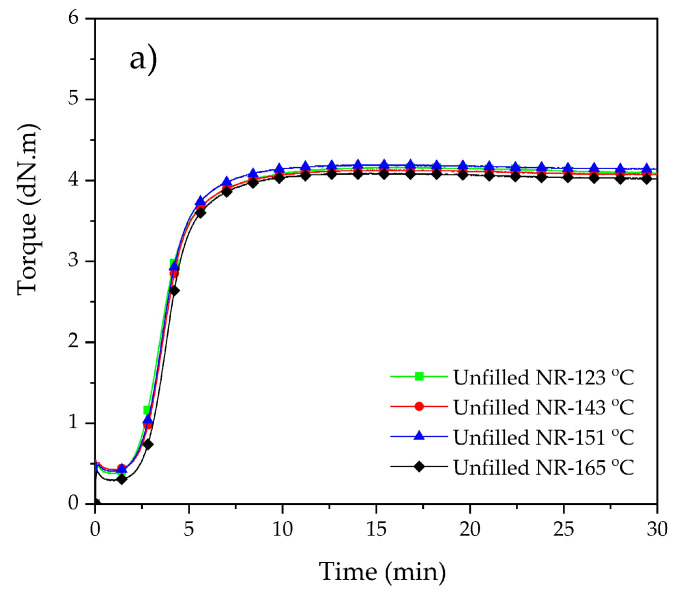
Cure characteristics of gum (**a**) and filled (**b**) NR compounds subjected to various dump temperatures.

**Figure 8 polymers-15-00160-f008:**
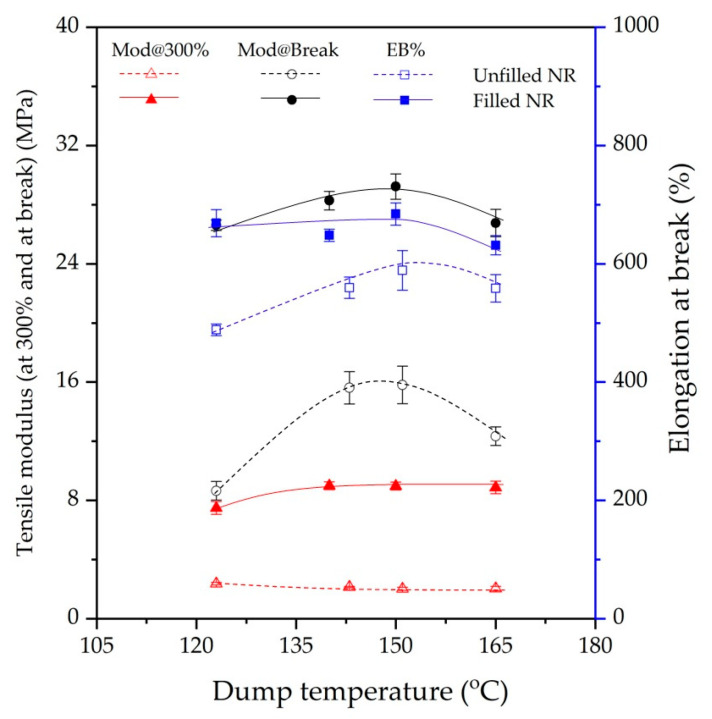
Tensile modulus (at 300% strain and at break) and elongation at break of gum and filled NR compounds subjected to various dump temperatures.

**Table 1 polymers-15-00160-t001:** Rubber formulations.

Ingredients	Amount (phr, Parts per 100 Rubber)		
	Masticated Pure NR	Gum NR	Filled NR
Natural Rubber (SMR10)	100.0	100.0	100.0
Silica (ULTRASIL 7005)	-	-	55.0
Silane (TESPD) ^a^	-	-	5.0
Process oil (TDAE)	-	-	8.0
Zinc oxide	-	3.0	3.0
Stearic acid	-	1.0	1.0
TMQ	-	1.0	1.0
6PPD	-	2.0	2.0
DPG ^a^	-	1.1	1.1
CBS	-	1.5	1.5
Sulfur	-	1.5	1.5

^a^ Amounts of TESPD and DPG were calculated according to the following equations [21]: TESPD (phr) = 0.00053 × Q × A; DPG (phr) = 0.00012 × Q × A, where Q is the amount of silica (phr) and A is the CTAB-specific surface area of the silica (171 m^2^/g).

**Table 2 polymers-15-00160-t002:** Mixing procedure.

Step 1: Non-Productive Masterbatch	
Time (min:sec)	Action
0:00	open ram; add rubbers
1:00	open ram; add ½ silica, silane
2:00	close ram
2:30	open ram; add ½ silica, oil, stearic acid, TMQ, 6PPD
3:30	sweep
4:30	close ram
6:45	dump
**Step 2: Productive compound**	
Time (min:sec)	Action
0:00	open ram; add masterbatch
1:00	open ram; add ZnO, DPG, CBS, and sulfur
2:00	close ram
5:00	dump

## Data Availability

Not applicable.

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
