# Peer review of "Dynamic Response and Molecular Chain Modifications Associated with Degradation during Mixing of Silica-Reinforced Natural Rubber Compounds"

_polymers, 2022, doi:10.3390/polym15010160_

Round 1

Reviewer 1 Report

The manuscript reports the dynamic response and molecular chain modifications associated with degradation during mixing of silica-reinforced natural rubber compounds, it is an interesting , well structured and well discussed work, but there are some observations that need to be done before, following there are detailed:

-it would be interesting to present a diagram of reaction of silica filled compounds, similar to presented in figure 3.

-For prepared formulation, which was the base for filler NR formulation? I mean amounts of TWM, 6 PPD, DPG, CBS, sulfur, Stearic acid and zinc oxide, how were established?

- For curves i figure 4, I recommend to use two Y-axis, I mean, in left side report G"and in right side G´, and use and arrow to indicate that correspond to each axis.

-in line 285, please write he complete word instead of abbreviation  for approximately.

-In experimental section indicate that the moduli at 100 and 300% strain were reported, but in results section only report moduli at 300%, please report value for 100% or correct in experimental section.

-In line 379, when indica that Tensile strength and elongation at break show an optimum value at 140-150ºC how establish that is the optimal value?

Author Response

Black color: comments of the reviewers

Blue color: our responses

Red color: modifications of the manuscript

Point 1: The manuscript reports the dynamic response and molecular chain modifications associated with degradation during mixing of silica-reinforced natural rubber compounds, it is an interesting, well structured and well discussed work, but there are some observations that need to be done before, following there are detailed:

Response 1: We thank the reviewer for the nice words.

Point 2: It would be interesting to present a diagram of reaction of silica filled compounds, similar to presented in figure 3.

Response 2: The reaction of silica in rubber compounds with help of silane coupling agent has been subject of numerous studies and been published by many authors. To name a few from our house:

- Reuvekamp, L.A.E.M.; ten Brinke, J.W.; van Swaaij, P.J.; Noordermeer, J.W.M. Effects of time and temperature on the reaction of TESPT silane coupling agent during mixing with silica filler and tire rubber, Rubber Chem. Technol. 2002, 75, 187-198.

- Kaewsakul, W.; Sahakaro, K.; Dierkes, W.K.; Noordermeer, J.W.M. Optimization of mixing conditions for silica-reinforced natural rubber tire tread compounds, Rubber Chem. Technol. 2012, 85, 277-294.

- Blume, A.; Jin, J.; Mahtabani, A.; He, X.; Kim, S.; Andrzejewska, Z. New structure proposal for silane modified silica. Kautsch. Gummi Kunstst. 2020, 5, 19-24.

And many, many more; e.g. refs. [2-4], [6-8], [15-18], [21] and [24] of the present manuscript.

In the present manuscript, we are focusing on the degradation reactions taking place on Natural Rubber primarily. In order not to confuse the latter with the mechanism of silanization and the effect it has on the properties, we prefer to keep these two mechanisms separate, for which reason we did not include it in the manuscript.

Point 3: For prepared formulation, which was the base for filler NR formulation? I mean amounts of TMQ, 6 PPD, DPG, CBS, sulfur, Stearic acid and zinc oxide, how were established?

Response 3: The filled NR formulation was based on truck tire tread compounds according to previous work of Kaewsakul et al., 2012; ref. [6]. The gum NR compound was derived therefrom and masticated pure NR stands on its own. We clarified that by inserting the following in the manuscript, paragraph 2.2:

The filled NR formulation was based on truck tire tread compounds according to previous work of Kaewsakul et al., 2012; ref. [6]. The gum NR compound was derived therefrom and the masticated pure NR was meant as reference for raw rubber.

Point 4: For curves in figure 4, I recommend to use two Y-axis, I mean, in left side report G" and in right side G´, and use an arrow to indicate that correspond to each axis.

Response 4: Implemented; we reported using two Y-axes in figure 4, the left side with G″ and the right side with G′.

Point 5: In line 287, please write the complete word instead of abbreviation for approximately.

Response 5: Implemented. The sentence is re-written to “The slight increase in moduli going from approximately 124°C to 140°C dump temperatures may be related to the onset of branch formation.”

Point 6: In experimental section indicate that the moduli at 100 and 300% strain were reported, but in results section only report moduli at 300%, please report value for 100% or correct in experimental section.

Response 6: Implemented; we took the 100% modulus out of the experimental section.

Point 7: In line 379, when indicate that tensile strength and elongation at break show an optimum value at 140-150ºC how establish that is the optimal value?

Response 7: In the previous work by Kaewsakul et al. ref. [6] about “Optimization of mixing conditions for silica-reinforced natural rubber tire tread compounds” it was reported that the optimal mixing conditions for the silica-filled NR compound are a dump temperature in the range of 135–150°C. The present work shows the same trends of tensile strength and elongation at break, that is the tensile strength and elongation at break reach maximum values at the temperature range of 140-150°C. Therefore, we mentioned the optimal values referring to this previous work: ref. [6]

Alternatively, please see the attachment.

Reviewer 2 Report

This is an interesting article dealing with the issue of NR modification in order to obtain better, expected properties, especially when used on car tires. The description of the material and methodology is basically  complete, as is the compilation of measurement results and their analysis.

However, before recommendation  that this article may be published, it is proposed to take into account the following comments and proposals:

·       Selection of mixture temperature; there are no information how the decision of the processing temperature was taken, have preliminary studies been carried out, or are these literature proposals? Did you find any relationship between the chamber temperature and the self-heating effect due to processing?

·       The contents of Table 1 are not clear; what were the real compositions of individual rubber compounds?

·       In lines 212 -213 it is noted that there has been a shortening of the macromolecule chains, which is a generally known phenomenon; can the authors provide proof (the result of measuring of the molecular mass distribution) for the specific rubber compounds discussed?

·       Line 214, please explain in details the proposed effect that a higher relaxation rate may be an indication of higher viscous/elastic rate?

·       Line 396-397  this conclusion is evident, even without any experimental work. Can you give any proof that really the crosslinking of shorten chains ahs taken place?

·       A general remark concerning the Conclusions; only the most important founding’s should be listed in this final part of the paper. In your case instead of Conclusions a summary of the results is presented with repeated values and discussion which may be found already at the text of this paper.

Author Response

Black color: comments of the reviewers

Blue color: our responses

Red color: modifications of the manuscript

Point 1: This is an interesting article dealing with the issue of NR modification in order to obtain better, expected properties, especially when used on car tires. The description of the material and methodology is basically complete, as is the compilation of measurement results and their analysis.

Response 1: We thank the reviewer for the nice words.

Point 2: Selection of mixture temperature; there are no information how the decision of the processing temperature was taken, have preliminary studies been carried out, or are these literature proposals? Did you find any relationship between the chamber temperature and the self-heating effect due to processing?

Response 2: The selection of the mixture temperatures or dump temperatures was made using higher and lower than optimal mixing conditions around 150°C dump temperature, based on the previous work of Kaewsakul et al., 2012; ref. [6]. It reported the deterioration of tensile properties when dump temperature exceeded 160°C. Then we confirmed these observations in a preliminary study with three conditions i.e. 120°C, 150°C and 165°C; later, we decided to add the additional condition of 140°C dump temperature.

We varied the mixer set temperatures and rotor speeds to achieve the required dump temperatures, measured by a hand-held thermocouple: see paragraph 2.2.3. The self-heating effect in the mixer was automatically compensated for by this approach.

Point 3: The contents of Table 1 are not clear; what were the real compositions of individual rubber compounds?

Response 3: Actually, the formulations are correct. The confusion of the reviewer may have come from the compound called “Masticated pure NR”, which was indeed pure NR without any other compounding ingredients.

Point 4: In lines 212 -213 it is noted that there has been a shortening of the macromolecule chains, which is a generally known phenomenon; can the authors provide proof (the result of measuring of the molecular mass distribution) for the specific rubber compounds discussed?

Response 4: We agree with the reviewer that shortening of macromolecular chains, or molecular weight decrease, under strenuous conditions is generally known. However to actually prove that with molecular mass distributions is difficult for elastomers in general, and practically impossible for natural rubber because of its secondary gel-like molecular structures. Ref. Y. Tanaka, “Structural characterization of natural polyisoprene: solve the mystery of natural rubber based on structural study”; Rubber chemistry and technology 74, 355-375 (2001). Therefore a lower Mooney viscosity is commonly associated with shortening of the rubber molecular chains.

Point 5: Line 214, please explain in details the proposed effect that a higher relaxation rate may be an indication of higher viscous/elastic rate?

Response 5: In polymer rheology stress relaxation is determined by a combination of both elastic and viscous responses, where a higher relaxation rate indicates an increase in viscous/elastic ratio: more liquid-like behavior than solid-like. Increase of this viscous/elastic ratio and so more liquid-like behavior is another sign of molecular mass decrease, i.e. a rising degree of degradation due to chain scission.

We added this in the manuscript with the same words:

In polymer rheology stress relaxation is determined by a combination of both elastic and viscous responses, where a higher relaxation rate indicates an increase in viscous/elastic ratio: more liquid-like behavior than solid-like. Increase of this viscous/elastic ratio and so more liquid-like behavior is another sign of molecular mass decrease, i.e. a rising degree of degradation due to chain scission.

Point 6: Line 396-397  this conclusion is evident, even without any experimental work. Can you give any proof that really the crosslinking of shorten chains are taken place?

Response 6: We agree with the reviewer that the particular sentence is rather evident. However, the novelty of the present study is the balance between chain scission, partially compensated by branch formation which has never been reported before. In the hope to explain this somewhat more clearly, we have changed the text as follows:

At too excessive mixing temperature, degradation by chain scission gives next to shorter chains also formation of branched structures, which partially compensate each other. However, overall this leads to an improper network structure and so inferior properties.

Point 7: A general remark concerning the Conclusions; only the most important founding’s should be listed in this final part of the paper. In your case instead of Conclusions a summary of the results is presented with repeated values and discussion which may be found already at the text of this paper.

Response 7: We shortened the conclusions as follows in the hope that it will satisfy the reviewer:

For pure NR and gum compounds, the changes in properties are mainly the result of chain scission or degradation, and a compensating amount of long-chain branch formation. The analysis of the long-chain branching with the Δδ gives supporting evidence for the generation of branched structures, as a result of increasing dump temperature. A different behavior is observed for the silica-filled NR compounds for which the properties show an overruling effect of the rubber-silica interactions/crosslinking on the elastic response. The changes of properties in relation to the dump temperatures demonstrate the existence of competitive reactions between the degradation of the pure rubber and crosslinking due to rubber-silica interaction. The latter involves both physical and chemical crosslinks promoted by silanization and coupling reactions. The chain modifications during mixing by either chain scission, chain recombination and long-chain branch formation clearly influence the mechanical properties of vulcanizates.  The overall results mark the intricate balance needed to be attained by the mixer operator between proper silanization with as little as possible degradation of NR in order to achieve optimal properties.

Alternatively, please see the attachment for our responses.

Reviewer 3 Report

This paper presents an analysis of the properties of silica-reinforced natural rubber compounds

Although the manuscript is arranged roughly factually correct, the research described herein does not bring significant novelty over the previously reported works. After getting acquainted with the work in depth, the Reviewer came to the conclusion that the results of the studied are very similar, and the experiments did not bring spectacular results. Perhaps, more samples and tests would make this article more attractive. In the introduction part, it should be included clear discussion of how this manuscript brings new information over papers that have already been published.

-        Authors should also consider the following aspects:

-        It is not possible for a work not to present microscopic (SEM) and macroscopic images of the samples before and after testing;

-         Only presenting graphs with the evolution of certain properties is not enough;

-        Conclusions should be more concrete and future research directions should be presented. It is not acceptable that much of the information in the abstract is taken over identically in the conclusions section.

Author Response

Black color: comments of the reviewers

Blue color: our responses

Red color: modifications of the manuscript

Point 1: This paper presents an analysis of the properties of silica-reinforced natural rubber compounds

Although the manuscript is arranged roughly factually correct, the research described herein does not bring significant novelty over the previously reported works. After getting acquainted with the work in depth, the Reviewer came to the conclusion that the results of the studied are very similar, and the experiments did not bring spectacular results. Perhaps, more samples and tests would make this article more attractive. In the introduction part, it should be included clear discussion of how this manuscript brings new information over papers that have already been published.

Response 1: Taking note of the comment of the reviewer, 10 days allowed for the reply is of course far too little to add new experiments. Of course, aging phenomena on natural rubber have been studied before. However, we would like to stress that the novelty of the present manuscript lies in the use of Δδ to register and quantify branch formation in natural rubber. The latter has never been employed before.

Point 2: Authors should also consider the following aspects:

- It is not possible for a work not to present microscopic (SEM) and macroscopic images of the samples before and after testing

Response 2: It is our experience that SEM and/or TEM are not very informative concerning aging phenomena of rubber. Morphological phenomena in rubber compounds can usually not be quantified in terms of aging, and the other way around. The dynamic properties do commonly clearly depend on aging phenomena. Therefore, we focused on the latter. And again, the Δδ employed to quantify branch formation is also a dynamic phenomenon, where this is a novelty in natural rubber research. We investigated the micro-dispersion indirectly as represented by the Payne effect versus various degrees of degradation: also a dynamic mechanical property.

Point 3: Only presenting graphs with the evolution of certain properties is not enough.

Response 3: We would like to refer the reviewer to our answer pertaining to lines 212-213 of the previous reviewer. It demonstrated the limitations of quantifying degradation of natural rubber in terms of molecular weight and molecular weight distribution. This work uses the Mooney stress relaxation together with the different phase angles as function of frequency sweep, both dynamic mechanical tests, to characterize long-chain branching and molecular weight distribution. This represents a novel approach, never been practiced before.

Point 4: Conclusions should be more concrete and future research directions should be presented. It is not acceptable that much of the information in the abstract is taken over identically in the conclusions section.

Response 4: The conclusions have been modified as per the answer to the second reviewer. We have added the following sentences to the Results and Discussion, because we think it better fits there:

The approach of future research is to investigate the effect of processing and aging stabilizers to prevent the degradation of natural rubber, which is more sensitive towards molecular breakdown and rearrangement than synthetic alternatives.

Alternatively, please see the attachment for our responses.

Round 2

Reviewer 3 Report

The authors did not respond to comment 1 and 2 respectively. It is not possible to present a paper based on experimental research and not present macroscopic and microscopic images for the tested specimens. The authors' refusal to present such images is not normal and it is possible that the results presented in the diagrams do not correspond to the real data.

Author Response

Point 1: The authors did not respond to comment 1 and 2 respectively. It is not possible to present a paper based on experimental research and not present macroscopic and microscopic images for the tested specimens. The authors' refusal to present such images is not normal and it is possible that the results presented in the diagrams do not correspond to the real data.

Response 1:  Thank you very much for your comment and suggestion. We had already answered your comments 1 & 2 with regard to the macroscopic and microscopic images conclusively in the previous round. 

Let us explain further for this context. 

Since the development of silica-technology for passenger tire tread rubbers it has been acknowledged that macroscopic and microscopic imaging does not lead to sensible results. Silica diffentiates itself completely from the more traditional carbon-black as reinforcing filler, for reason that the latter interacts with rubber on basis of van der Waals interactions, no more, while silica interaction with rubber needs to  be achieved with rubber by chemial coupling via silane coupling agents. Forces at least 10 times higher than van der Waals’s. For carbon black, because of its very low interaction forces, the dispersion in the rubber matrix may in some cases be poor, which can be seen by large agglomerates (several µm’s) in rubber compounds visible by the naked eye or with optical microscopy. The silica’s for rubber reinforcement are manufactured on purpose to disperse easily to the level of primary particles (several 10’s of nm) by the so-called micro-pearl process. The main issue with silica-reinforcement is then the generation of the chemical silanization reaction, the extent of which is NOT visible with microscopy but quantified with secondary methods like dynamic and mechanical properties before and after vulcanisation. As we did.  

For that reason we did not include microscopic details, because it diverts the reader from the essence of the manuscript and it does not add value to the present manuscript, which focuses on ageing phenomena.